# Homo- and hetero-dimeric subunit interactions set affinity and efficacy in metabotropic glutamate receptors

Chris Habrian[1,6], Naomi Latorraca [2], Zhu Fu[2] & Ehud Y. Isacoff [1,2,3,4,5] ✉

Metabotropic glutamate receptors (mGluRs) are dimeric class C G-protein–coupled receptors that operate in glia and neurons. Glutamate affinity and efficacy vary greatly between the eight mGluRs. The molecular basis of this diversity is not understood. We used single-molecule fluorescence energy transfer to monitor the structural rearrangements of activation in the mGluR ligand binding domain (LBD). In saturating glutamate, group II homodimers fully occupy the activated LBD conformation (full efficacy) but homodimers of group III mGluRs do not. Strikingly, the reduced efficacy of Group III homodimers does not arise from differences in the glutamate binding pocket but, instead, from interactions within the extracellular dimerization interface that impede active state occupancy. By contrast, the functionally boosted mGluR II/III heterodimers lack these interface 'brakes' to activation and heterodimer asymmetry in the flexibility of a disulfide loop connecting LBDs greatly favors occupancy of the activated conformation. Our results suggest that dimerization interface interactions generate substantial functional diversity by differentially stabilizing the activated conformation. This diversity may optimize mGluR responsiveness for the distinct spatio-temporal profiles of synaptic *versus* extrasynaptic glutamate.

G-protein-coupled receptors (GPCRs) constitute the largest class of membrane signaling proteins. GPCRs respond to a wide array of extracellular stimuli to initiate intracellular signaling[1]. Recent studies have revealed distinct conformations[2–5] and dynamics[6–12], which are associated with ligand recognition, activation, and signaling[9,13,14], including in monomeric class A receptors for neuromodulators and dimeric class C GPCRs for the excitatory neurotransmitter glutamate, the metabotropic glutamate receptors (mGluRs)[15–17]. mGluRs regulate neurotransmitter release and neuronal excitability, functions that are critical to learning and memory[18,19]. mGluR mis-regulation is implicated in psychosis and cognitive impairment[20,21]. The mammalian brain has eight mGluR subtypes, including group I members mGluR1 and 5, which signal through $G_q$, and group II members mGluR2 and 3, and

group III members mGluR4,6,7, and 8, which are often presynaptic and signal through $G_i$[22]. Apparent glutamate affinity varies over 4 orders of magnitude across the mGluRs, from micromolar to millimolar, and glutamate efficacy (the fraction of maximal activation achieved at a saturating glutamate concentration) varies over 20-fold, from 5 to 100%[15,16]. Apparent affinity and efficacy are powerfully modulated by certain heterodimeric combinations[16] and activity is elevated by trans-synaptic interaction with ELFN proteins[23]. The molecular mechanisms underlying both function differentiation and regulation are not well understood.

To elucidate the molecular determinants that control mGluR affinity and efficacy, we used fluorescence resonance energy transfer (FRET) between donor and acceptor fluorophores attached to a SNAP

[1]Biophysics Graduate Group, University of California, Berkeley, CA, USA. [2]Department of Molecular and Cell Biology, University of California, Berkeley, CA, USA. [3]Helen Wills Neuroscience Institute, University of California, Berkeley, CA, USA. [4]Weill Neurohub, University of California, Berkeley, CA, USA. [5]Molecular Biology & Integrated Bioimaging Division, Lawrence Berkeley National Laboratory, Berkeley, CA, USA. [6]Present address: Department of Molecular and Cellular Physiology, Stanford University School of Medicine, Stanford, CA, USA. ✉e-mail: ehud@berkeley.edu

domain fused to the N-terminal end of the ligand binding domain (LBD) of each subunit of the mGluR dimer to measure the molecular motions that change the relative positions of the LBDs as receptors transition between inactive and active states[24,25]. Using single molecule FRET (smFRET) in total internal reflection fluorescence microscopy, we measured conformational dynamics in isolated receptor dimers displayed at low density on a passivated coverslip[12,15–17,26,27].

Consistent with earlier work[12,15–17], we found that in saturating glutamate group II members fully occupy the activated conformation, whereas group III members do not. Homodimeric receptors with domain swaps between high apparent affinity/high efficacy group II member mGluR2 and low apparent affinity / low efficacy group III member mGluR7 revealed a strong influence on these properties of the LBD and cysteine rich domain (CRD), but not of the transmembrane domain (TMD). Strikingly, differences in apparent affinity and efficacy were found to arise not from differences in the glutamate binding pocket but rather from differences in the relative occupancy of conformations along the activation pathway, which tilt more toward the activated state in group II than in group III homodimers and most in group II/III heterodimers. Our experiments suggest that these differential stabilities arise from differences in subunit interaction at three interfaces: the lower LBD lobe interface and the CRD interface, both of which come into contact in only the active conformation, and the cysteine loop, which forms a disulfide link between upper LBD lobes. Our findings suggest that dimer interactions between lower LBD lobes and between CRDs prevent homodimeric group III mGluRs from fully activating and that this dual brake is relieved by II/III cysteine loop interaction. This multi-point subunit-to-subunit communication provides a mechanism for receptor tuning that enables mGluRs to function over the wide range of concentration, temporal and spatial scales found in synaptic communication.

## Results

### Incomplete active state occupancy across Group III mGluRs

To understand the mechanism that generates diversity in glutamate affinity and efficacy across mGluRs, we first set out to confirm that the previously described differences between glutamate group II member mGluR2 (high affinity/high efficacy) and group III member mGluR7 (low affinity/low efficacy)[16]. We examined the glutamate dependence of occupancy of the activated conformation of the Venus Fly Trap (VFT) ligand binding domain (LBD) using inter-LBD single molecule fluorescence energy transfer (smFRET) between donor and acceptor

dyes attached to SNAP tags fused to the N termini of the two subunits of the receptor, at the "top" of the LBDs (Supplementary Fig. 1a, b)[10,12,24,25]. Glutamate binding stabilizes a closed conformation of the LBD and LBD reorientation that increases the distance between LBD N-termini, reducing FRET (Supplementary Fig. 1c). These LBD rearrangements bring the cysteine rich domains (CRDs) and transmembrane domains (TMDs) into contact and present binding sites for G protein on the inner surface of the TMDs, resulting in signaling (Supplementary Fig. 1d)[28–30].

HEK293T cells expressing SNAP-mGluR7 or SNAP-mGluR2 were labeled with a mixture of Alexa-647 (acceptor) and DY-547 (donor) fluorophores (Förster radius = 52 Å). The cells were lysed in detergent to solubilize the receptors, then immune-purified and tethered by biotinylated secondary antibodies at low density to coverslips that were passivated with polyethylene glycol. We used total internal reflection fluorescence (TIRF) microscopy to detect single fluorophores for smFRET and measured the absolute FRET levels and dynamic changes associated with ligand-induced conformational changes in the LBDs.

In zero glutamate, homodimeric SNAP-mGluR2 (mGluR2/2) had a tight FRET distribution with a single peak at a FRET value -0.45 (Fig. 1a, black). At an intermediate glutamate concentration of 10 μM glutamate, we observed a bimodal distribution with about 30% occupancy in the high FRET (-0.45) resting conformation and 70% occupancy of the low FRET (-0.2) activated conformation (Fig. 1a, blue). At a saturating concentration of 100 mM glutamate, there was complete occupancy of the low FRET activated state (-0.2) (Fig. 1a, red), as had been shown before even at the much lower concentration of 1 mM glutamate in both mGluR2 and mGluR3[12,16], confirming that group II mGluRs have 100% efficacy. In contrast to mGluR2/2, although mGluR7/7 had a tight FRET distribution with a single peak at a high FRET resting state value of -0.5 in zero glutamate (Fig. 1b, black), at 100 mM glutamate, the low FRET activated state (-0.2) appeared but had very low (-5%) occupancy (Fig. 1b, red), reflecting a very low efficacy, as described before[16].

To determine if incomplete occupancy of the LBD activated conformation in saturating glutamate is unique to mGluR7, we examined two other group III members that are expressed broadly in the brain, mGluR4 and mGluR8. In zero glutamate, the mGluR4/4 homodimer had a single high FRET peak (-0.45) (Fig. 1c, black), similar to the resting LBD conformation of mGluR7 and mGluR2. At an intermediate glutamate concentration of 10 μM, mGluR4/4 had a bimodal distribution that was approximately evenly divided between the high FRET

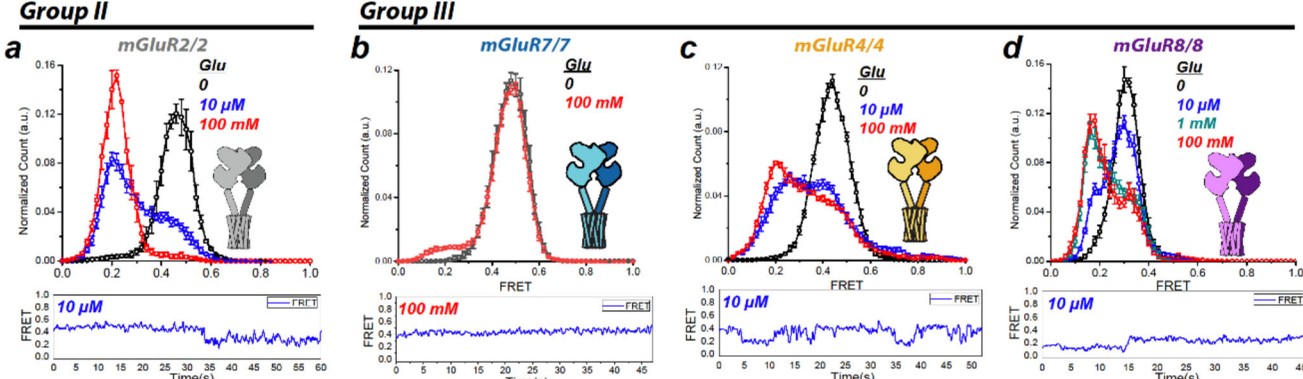

**Fig. 1 | Glutamate-induced ligand binding domain conformational changes of Group II and III mGluR homodimers.** smFRET distributions (top, smFRET values mean ± sem; N = number of molecules) and representative trace of single dimer (bottom) at different glutamate concentrations in four mGluR homodimers from two groups: **a** SNAP-mGluR2 in glutamate (0: N = 240, 10 μM: N = 258, 68% active, 100 mM: N = 236, 129% active). **b** SNAP-mGluR7 in glutamate (0: N = 252, 100 mM: N = 231, 5% active). **c** SNAP-mGluR4 in glutamate (0: N = 266, 10 μM: N = 217, 45% active, 100 mM: N = 263, 51% active). **d** SNAP-mGluR8 in glutamate (0: N = 330, 10 μM: N = 340, 33% active, 1 mM: N = 295, 76% active, 100 mM: N = 286, 76% active). Donor (BG-DY-547) and acceptor (BG-Alexa-647) dyes imaged at 10 fps. Maximal activation (low FRET peak/low FRET peak + high FRET peak): mGluR2/2 = 100%; mGluR 4/4 = 65%, mGluR7/7 = 5%; mGluR8/8 = 80%.

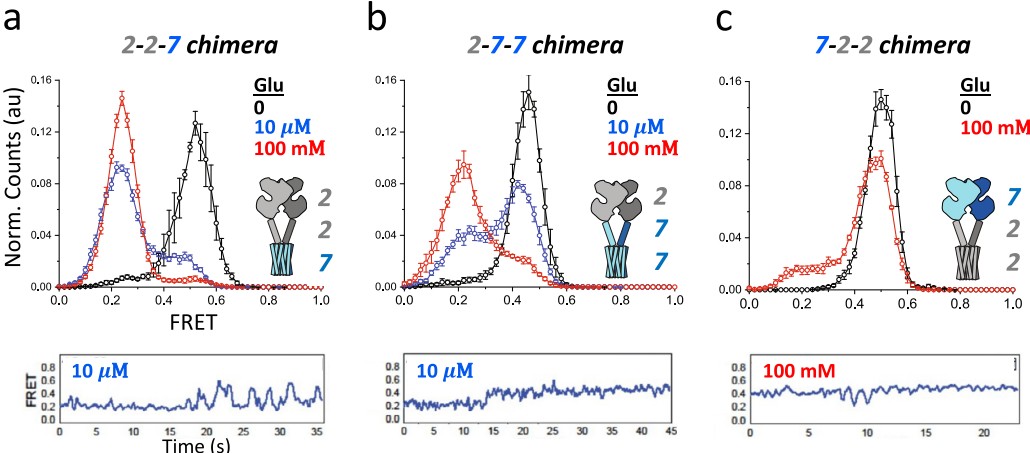

**Fig. 2 | Domain swaps between mGluR2 and mGluR7 reveal the influence of LBD and CRD on ligand efficacy.** smFRET distributions (top, values mean ± sem; N = number of molecules) and representative trace of single dimer (bottom) from chimeric receptors between mGluR2 and mGluR7 made by exchanges LBD, CRD, TMD, as indicated in cartoon insets, with representative trace of single dimer (bottom): **a** SNAP-mGluR2(LBD + CRD)-mGluR7(TMD) in glutamate (0: N = 180, 10 μM: N = 242, 75% active, 100 mM: N = 293, 120% active). **b** SNAP-mGluR2(LBD)-mGluR7(CRD + TMD) in glutamate (0: N = 181, 10 μM: N = 198, 27% active, 100 mM: N = 233, 63% active). **c** SNAP-mGluR7(LBD)-mGluR2(CRD + TMD). in glutamate (0: N = 199, 100 mM: N = 266, 12% active).

resting conformation (-0.45) and low FRET activated conformation (-0.2) (Fig. 1c, blue), near the EC50. In 100 mM glutamate, the low FRET activated conformation (-0.2) had an occupancy of ~60%, (Fig. 1c, red). This shows that the maximal efficacy of mGluR4 is limited, although not as severely as in mGluR7. We next turned to mGluR8/8. In zero glutamate, we observed a single narrow distribution at ~0.35 FRET (Fig. 1d, black), and in 10 μM glutamate a bimodal distribution with ~30% occupancy of low (-0.2) FRET active conformation (Fig. 1d, blue). Both 1 mM and 100 mM glutamate resulted in a bimodal distribution with ~65% occupancy of the active conformation (Fig. 1d, 1 mM teal; 100 mM red), another case of inability to fully occupy the activated conformation in saturating glutamate.

Together, these observations indicate that glutamate fully stabilizes the active conformation of group II mGluRs but only partly for group III mGluRs. Although the mGluR with the lowest efficacy, mGluR7/7, also had the lowest apparent affinity, these properties were not strictly associated, as seen, for example, from the fact that mGluR4/4 had a higher apparent affinity but lower efficacy than mGluR2/2 (Fig. 1e).

## The LBD and CRD set glutamate activation

To identify the molecular determinants of mGluR activation, we swapped the three mGluR domains--the LBD, CRD, and TMD−between the high apparent affinity/high efficacy mGluR2 and the low apparent affinity/low efficacy mGluR7 and studied these as homodimers. The LBD activation rearrangement of the chimera containing the LBD and CRD of mGluR2 and TMD of mGluR7 (2-2-7) closely resembled that of mGluR2/2, with full occupancy of the resting (-0.5) high FRET conformation in zero glutamate (Fig. 2a, black), full occupancy of the activated low FRET ( ‑ 0.2) conformation in 100 mM glutamate (Fig. 2a, red), and ~ 70% occupancy of the activated conformation in 10 μM glutamate (Fig. 2a, blue). This suggests that the TMD of mGluR7 has little influence on affinity and efficacy. We next tested a construct with the mGluR2 LBD and the CRD and TMD of mGluR7 (2-7-7). 2-7-7 had full occupancy of the (-0.5) high FRET resting conformation in zero glutamate (Fig. 2b, black), ~40% occupancy of the active state in 10 μM glutamate (Fig. 2b. blue) and ~80% occupancy of the low FRET activated conformation at 100 mM glutamate (Fig. 2b, red), a modest reduction in apparent affinity and efficacy. A construct containing the mGluR7 LBD followed by the mGluR2 CRD and TMD (7-2-2) went from full occupancy of the high FRET resting conformation (-0.5) in zero glutamate (Fig. 2c, black) to very low (-10%) occupancy of the low FRET

activated conformation at 100 mM glutamate (Fig. 2c, red), closely resembling mGluR7. Thus, the LBD has the dominant effect on efficacy and the CRD a smaller effect.

## Lower LBD interface a key determinant of group III mGluR low efficacy

The mGluR LBDs differ in amino acid sequence throughout, including in the glutamate binding pocket. To map the determinants of affinity and efficacy, we made chimeras from two group III mGluRs, mGluR7, and mGluR4, which are relatively closely related in amino acid sequence (70% homology/42% identity) (Supplementary Fig. 2a) but differ greatly in efficacy (by ~12-fold) (Fig. 1b, c) and apparent affinity[16]. The conservation between the sequences is unevenly distributed in the LBD (Supplementary Fig. 2a). We focused on the first 340 amino acids of the VFT comprising the LBD interface of the upper lobe (amino acids 40−202) and LBD interface of the lower lobe (amino acids 203−340) and ligand interacting residues (Supplementary Fig. 2b). We first tested a chimera that transplanted both LBD upper lobe interface and LBD lower lobe interface (residues 40−340) of mGluR4 into mGluR7. mGluR7(mGluR4: LBD 40−340) showed ~76% occupancy of the activated low FRET state at 4 μM glutamate (Fig. 3a, blue) and ~85% occupancy of the activated state at 1 mM and 10 mM glutamate (Fig. 3a, green and red), indicating that 85% represents maximal efficacy. Thus, together, the upper interface (Supplementary Fig. 2b purple) the lower interface (Supplementary Fig. 2b, teal) of the mGluR4 LBD endow the high apparent affinity and elevated efficacy of mGluR4 (Fig. 1c).

Only two of the twelve residues that compose the glutamate binding pocket−residue 74 in the upper lobe and residue 287 in the lower lobe−differ between mGluR4 and mGluR7 (Supplementary Fig. 2a), making these prime candidates for setting differences in affinity and efficacy between these receptor subtypes. Along with these, nearby residue 258 also differs between mGluR4 and mGluR7. To test the role of these residues, we mutated these residues in mGluR7 as a group to their mGluR4 identities (N74K, Q258R, D287E). Strikingly, the glutamate activation of mGluR7(K74, R258, E287) was indistinguishable from that of wildtype mGluR7, with very low efficacy (-5%) at 10 mM (Fig. 3b, red) glutamate and almost no activation at 1 mM glutamate (Fig. 3b, green). This suggests that differential glutamate coordination in the orthosteric binding pocket is not responsible for the pronounced differences in apparent affinity and efficacy between mGluR4 and mGluR7.

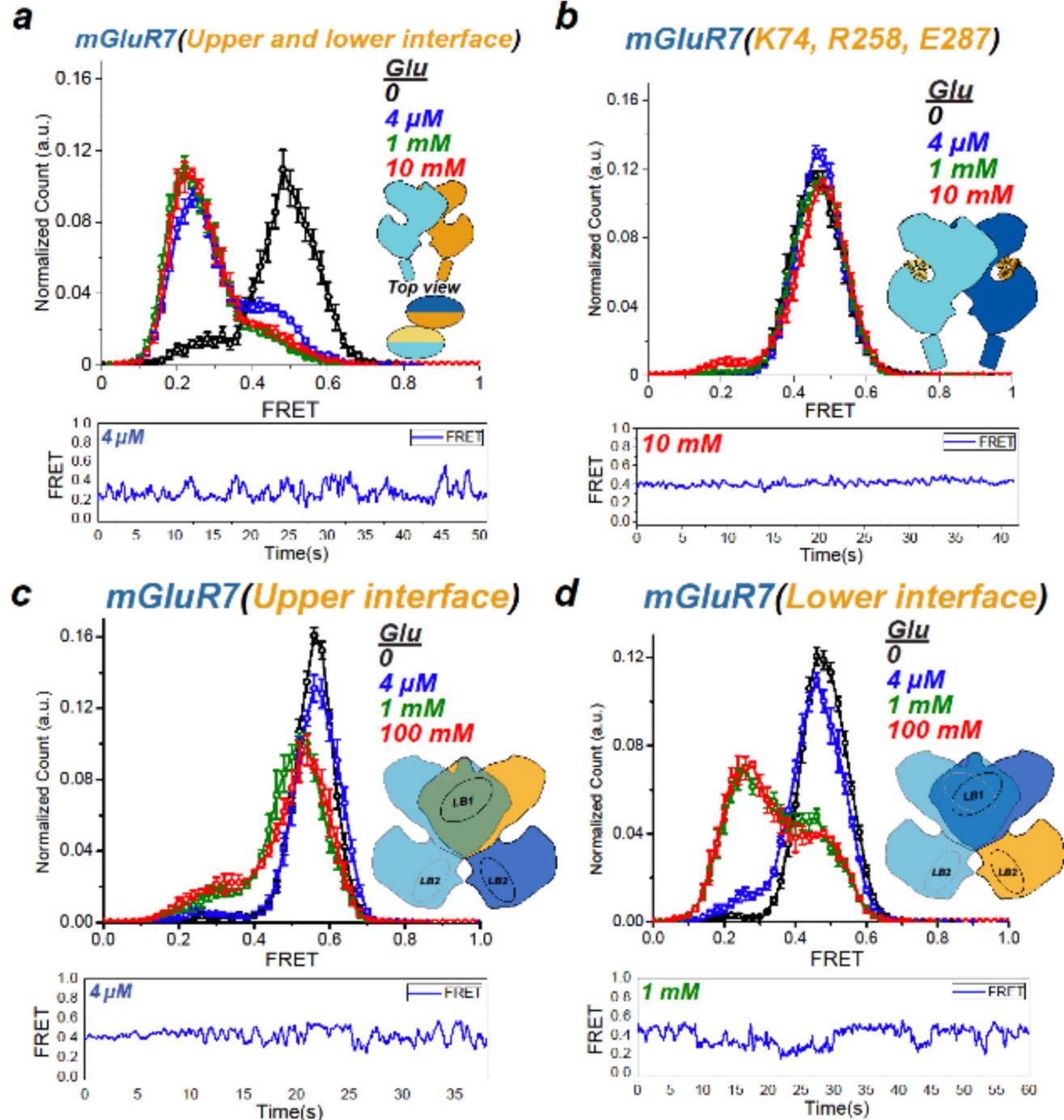

**Fig. 3 | The LBD dimer interface determines ligand efficacy.** smFRET distributions (top, values mean ± sem.; *N* = number of molecules) and representative trace of single dimer (bottom) from chimeric receptors between mGluR2 and mGluR7 made by exchanges of portions of the LBD, as indicated in cartoon insets. **a** SNAP-mGluR7(mGluR4 residues 40–340) in glutamate (0: *N* = 226, 8% active 4 μM: *N* = 320, 72% active, 1 mM: *N* = 459, 100% active, 100 mM: *N* = 429, 100% active). **b** SNAP-mGluR7(mGluR4 residues 74, 258 and 287) in glutamate (0: *N* = 356, 4 μM: *N* = 259, 1 mM: *N* = 339, 100 mM: *N* = 313, 5% active). **c** SNAP-mGluR7(mGluR4 residues 40–202) in glutamate (0: *N* = 231, 4 μM: *N* = 296, 2% active, 1 mM: *N* = 319, 10% active, 100 mM: *N* = 306, 10% active). **d** SNAP-mGluR7(mGluR4 residues 203–340) in glutamate (0: *N* = 328, 4 μM: *N* = 372, 8% active, 1 mM: *N* = 395, 58% active, 100 mM: *N* = 367, 58% active).

To narrow down the molecular components that are responsible for the difference in apparent affinity and efficacy between mGluR4 and mGluR7, we separately transplanted either the upper LBD interface or the lower LBD interface from mGluR4 into mGluR7. The upper LBD interface swap [mGluR7(mGluR4: 40–202)] reached saturating occupancy of the low FRET activated state at ~20% (Fig. 3c). The lower LBD interface swap [mGluR7(mGluR4: 203–340)] reached saturating occupancy of the low FRET activated state at ~70% (Fig. 3d). Thus, the

upper LBD interface and lower LBD interface of mGluR4 each increase the apparent affinity and efficacy of mGluR7, with the lower LBD interface having the bigger effect.

The influence of the upper and lower LBD interfaces suggests that interactions between subunits may influence affinity and efficacy by differentially stabilizing the resting and active conformations. The upper LBD dimer interface interacts in both the resting and active conformations but changes its contacts, whereas the lower LBD dimer

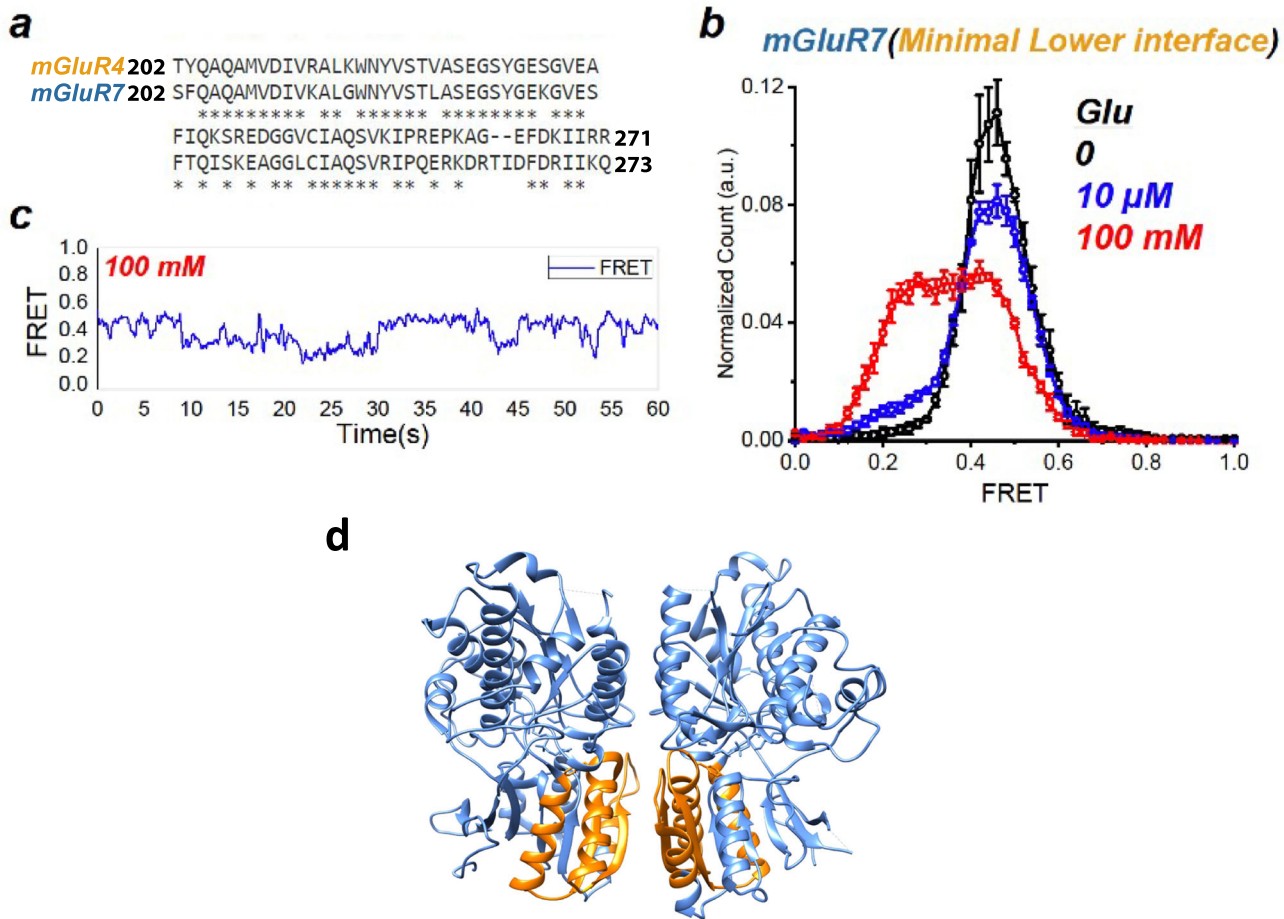

**Fig. 4 | Minimal LBD lower lobe mGluR4 interface required to increase mGluR7 ligand efficacy. a** LBD lower lobe sequence alignment between mGluR4 and mGluR7 (identical sites indicated by asterisks). **b** smFRET distributions of a minimal chimera in which a 70 amino acid stretch of the mGluR4 lower lobe is swapped into mGluR7: SNAP-mGluR7(mGluR4 residues 202–271) in glutamate (0: $N = 213$, 10 μM: $N = 313$, 12% active, 100 mM: $N = 288$, 55% active) Values mean ± sem; $N$ = number of molecules. **c** Representative trace of SNAP-mGluR7(mGluR4 residues 202–271) in 100 mM glutamate, which **b** shows to occupy the active (low FRET) conformation approximately half of the time. **d** Model of dimeric LBD (light blue) highlighting interface residues 202–271 (orange) on the active state structure of the mGluR4/4 homodimer (PDB 7E9H[37]).

interface interacts only in the active conformation[28–30]. Because of its larger influence, we focused the lower LBD interface, and replaced residues 202–273 of mGluR7 with residues 202–271 of mGluR4 [mGluR7(mGluR4: 202–273)]. This region contains 48 residues that are identical between mGluR4 and mGluR7, 9 residues that are similar, 12 residues with non-conservative differences, and a 2-residue insert in mGluR7 (Fig. 4a). In zero glutamate, mGluR7(mGluR4: 202–273) fully occupied the high FRET resting LBD conformation (Fig. 4b, black). In 10 μM glutamate, there was ~10% occupancy of the active conformation (Fig. 4b, blue) and in 100 mM glutamate occupancy shifted to ~50% of the low FRET activated conformation (Fig. 4b, red). This behavior is similar to that of the full lower LBD swap (Fig. 3d) and of wildtype mGluR4 (Fig. 1d). These observations suggest that the LBD lower lobe dimer interface differentially destabilizes the active conformation of group III mGluRs to limit to various degrees their efficacy and apparent affinity. We also tested three smaller regions of mGluR4 contained within the ~70 amino acid region of the minimal interface and found that none alone had a measurable effect (Supp. Figure 3), suggesting that a network of interactions sets glutamate efficacy.

## CRD role in efficacy

As seen with the lower LBD dimer interface, the CRD also forms a dimer interface exclusively in the activated state[28–30]. Moreover, while differences in efficacy are associated with differences in the LBD

(Figs. 2–4), the CRD also contributes, as seen from domain swaps between mGluR2 and mGluR7 (Fig. 2a, blue versus Fig. 2b, blue). To focus specifically on efficacy, we extended our analysis to CRD swaps between mGluR2 and mGluR4, which differ in efficacy, but have similar high apparent affinity (Fig. 1a, c, e). We find that mGluR2 with the mGluR4 CRD [mGluR2(mGluR4 CRD)] only reaches ~40% occupancy of the low FRET activated state in 100 mM glutamate (Fig. 5a, red), far below the full occupancy seen in mGluR2 (Fig. 1a, red). By the same token, mGluR4 with the mGluR2 CRD [mGluR4(mGluR2 CRD)] reached ~80% occupancy of the low FRET value activated state in 100 mM glutamate (Fig. 5b, red), intermediate between what is seen in mGluR2 and mGluR4 (Fig. 1b, red and Fig. 1d, red). The ability of the mGluR2 CRD to increase efficacy in mGluR4 and of the mGluR4 CRD to decrease the efficacy of mGluR2 suggests that the Group II CRD confers a higher maximal efficacy than does the Group III CRD.

## Cysteine loop contributes to heterodimer modulation

Previously, we showed that heterodimerization between a group II member mGluR2 or mGluR3 and group III member mGluR7 greatly boosts both the apparent affinity and efficacy of mGluR7[16]. We wondered whether the dimer interface determinants that control affinity and efficacy in group III homodimers play a role in this II/III heterodimer boost. To address this, we compared the LBD-LBD conformational dynamics of mGluR2/7 to those of mGluR2/4, which Pin and

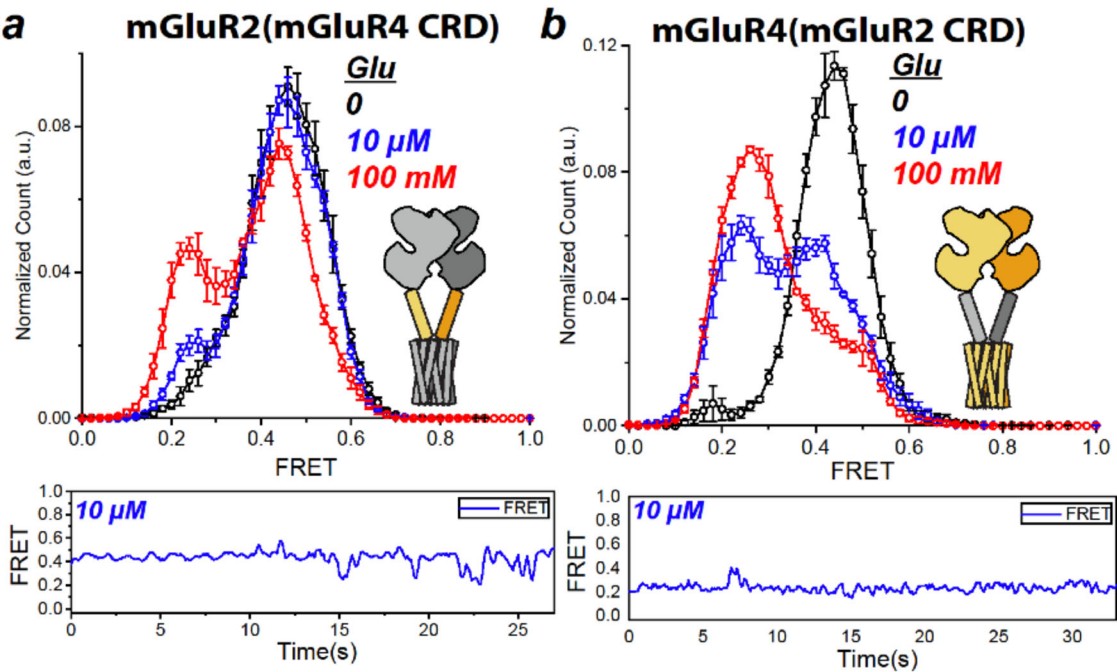

**Fig. 5 | Cysteine rich domain modulates ligand efficacy.** smFRET distributions (top, values mean ± sem; $N$ = number of molecules) and representative trace of single dimer (bottom) from chimera of mGluR2 with the CRD from mGluR4 and chimera of mGluR4 with the CRD from mGluR2 as indicated in cartoon insets, with representative trace of single dimer (bottom): **a** SNAP-mGluR2(mGluR4 CRD) in glutamate (0: $N$ = 306, 10 μM: $N$ = 392, 20% active, 100 mM: $N$ = 386, 48% active). **b** SNAP-mGluR2(mGluR4 CRD) in glutamate (0: $N$ = 247, 10 μM: $N$ = 250, 53% active, 100 mM: $N$ = 314, 79% active).

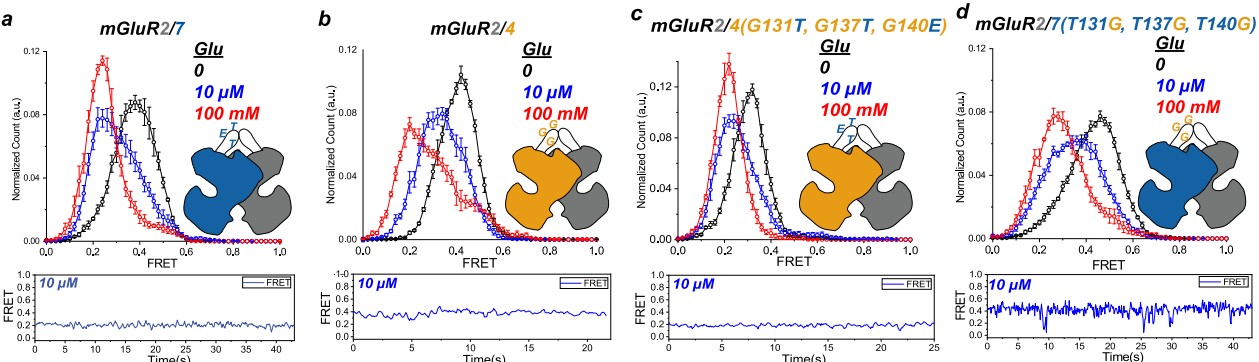

**Fig. 6 | Three cysteine loop residues enable heterodimeric positive modulation between LBDs.** smFRET distributions (top, smFRET values mean ± sem; $N$ = number of molecules) and representative trace of single dimer (bottom) for: CLIP-mGluR2/SNAP-mGluR7 and CLIP-mGluR2/SNAP-mGluR4 and the same heterodimers where the cysteine loops of mGluR4 and mGluR7 have three point mutations. **a** CLIP-mGluR2/SNAP-mGluR7 in glutamate (0: $N$ = 361, 10 μM: $N$ = 305, 88% active, 100 mM: N = 292, 140% active. **b** CLIP-mGluR2/SNAP-mGluR4 in glutamate (0: $N$ = 246, 10 μM: $N$ = 183, 37% active, 100 mM: $N$ = 143, 71% active) Values mean ± sem. **c** CLIP-mGluR2/SNAP-mGluR4(G131T, G137T, G140E) in glutamate (0: $N$ = 229, 10 μM: $N$ = 189, 82% active, 100 mM: $N$ = 179, 117% active). **d** CLIP-mGluR2/SNAP-mGluR7(T131G, T137G, E140G) in glutamate (0: $N$ = 254, 10 μM: $N$ = 260, 58% active, 100 mM: $N$ = 213, 75% active).

colleagues found to have a unique pharmacology[24,31,32]. Site-specific labeling of heterodimeric samples was achieved using a CLIP tag attached to mGluR2 and a SNAP tag attached to mGluR7 or mGluR4. In zero glutamate, mGluR2/7 had a broad distribution at high FRET (~0.4) (Fig. 6a, black). In stark contrast to the 36 mM EC50 and 5% maximal occupancy of the active conformation seen in the mGluR7/7 homodimer (Fig. 1c), mGluR2/7 shifted to ~90% occupancy of the low FRET (~ 0.25) activated conformation in 10 μM glutamate (Fig. 6a, blue), and almost full occupancy seen in 100 mM glutamate (Fig. 6a, red), agreeing with earlier observations[16]. mGluR2/4 resembled mGluR2/7 in zero glutamate, with a single broad FRET distribution centered at ~0.45 (Fig. 6b, black). However, in 10 μM glutamate, the FRET distribution

broadened and shifted to an intermediate peak value of ~0.35 (Fig. 6b, blue) and in 100 mM glutamate, the distribution was even broader, encompassing values between the ~0.25 FRET active conformation, the ~0.35 FRET intermediate peak and the distribution seen in zero glutamate with a peak at 0.45 FRET (Fig. 6b, red). This behavior was similar to that of the mGluR4/4 homodimer (Fig. 1d), suggesting that there is little or no heterodimeric boost in GluR2/4.

To understand the molecular mechanism of the heteromeric boost in mGluR2/7, we endeavored to identify the molecular determinants that mGluR7 possesses and mGluR4 lacks by transplanting components of mGluR7 into mGluR4 and focused on the cysteine loop. We focused on the cysteine loop, a flexible segment at the top of

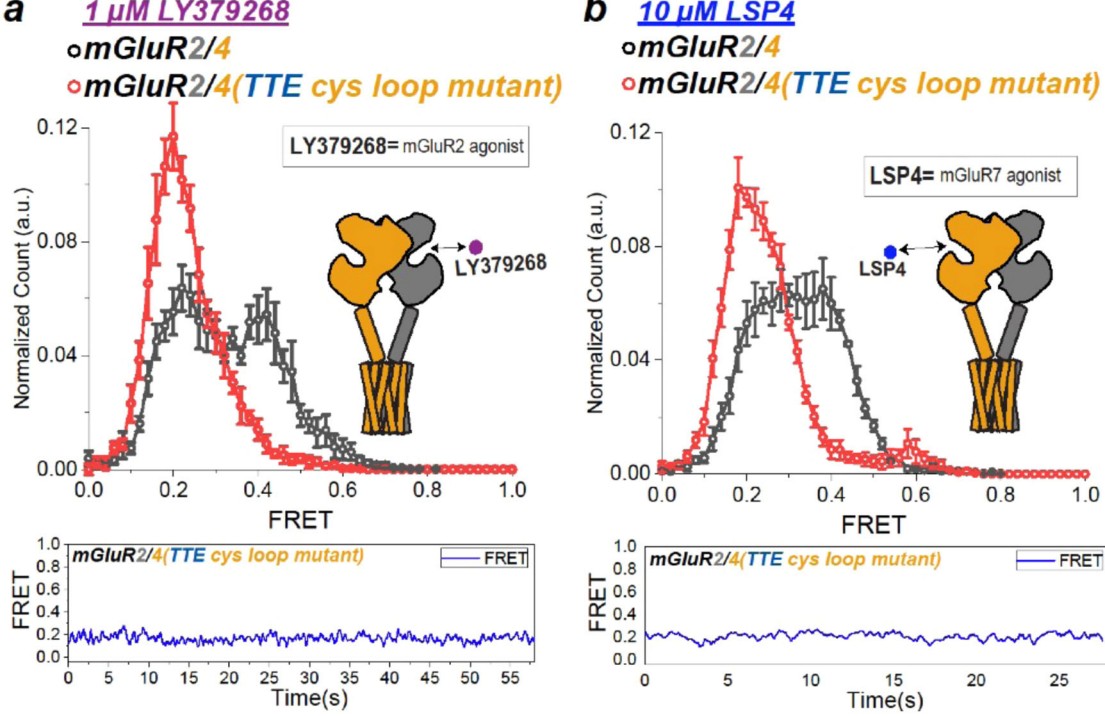

**Fig. 7 | Three cysteine loop residues from mGluR7 enable the closure of empty LBD in heterodimeric mGluR2/4.** smFRET distributions (top, smFRET values mean ± sem; $N$ = number of molecules) and representative trace of single dimer (bottom): CLIP-mGluR2/SNAP-mGluR4 (black) and CLIP-mGluR2/SNAP4(G131T, G137T, G140E) (red) in saturating in 1 uM LY379268 (Group II mGluR specific agonist) and 10 uM LSP4-2022 (Group III specific agonist). **a** CLIP-mGluR2/SNAP-mGluR4 (black, $N$ = 150, 59% active) and CLIP-mGluR2/SNAPmGluR4(G131T, G137T, G140E) mutant (red, $N$ = 143, 127% active) in saturating in 1 μM LY379268 (Group II specific agonist). **b** CLIP-mGluR2/SNAP-mGluR4 (black, $N$ = 198, 52% active) and CLIP-mGluR2/SNAPmGluR4(G131T, G137T, G140E) (red, $N$ = 174, 112% active) in saturating in 10 μM LSP4-2022 (Group III specific agonist).

the LBD that makes an inter-subunit cysteine bridge[33,34]. One aspect of the cysteine loop sequence which stands out is that in mGluR7 it contains one glycine residue, which is conserved in mGluR4, whereas in mGluR4 it contains three glycines that are not found in mGluR7 (Supp. Fig. 4). We mutated the three unique glycines of mGluR4 to the identity of their counterparts in mGluR7: G131T, G137T and G140E. Strikingly, in 10 µM glutamate, mGluR2/4(G131T, G137T, G140E) had a high occupancy of the low FRET (~0.2) activated conformation (Fig. 6c, blue) and in 100 mM glutamate there was full occupancy of the activated conformation (Fig. 6c, red). Moreover, the zero glutamate FRET distribution of mGluR2/4(G131T, G137T, G140E) (Fig. 6c, black) was left-shifted with respect to that of mGluR2/4 (Fig. 6b, black), as well as that of the mGluR2/2 and mGluR7/7 homodimers (Fig. 1a, b), but similar to that of mGluR2/7 (Fig. 6a, black), which we showed earlier to reflect partial activation in the Apo state[16]. These behaviors indicate that the triple mutation is sufficient to endow mGluR4 with the heterodimer boost. We also tested the reverse cysteine loop cysteine swap, mGluR7(T131G, T137G, E140G), in which the mGluR4 three glycines were substituted into mGluR7. mGluR2/7(T131G, T137G, E140G) (Fig. 6d) behaved similarly to mGluR2/4 (Fig. 6b), indicating that the TTE motif in the mGluR7 cysteine loop is necessary for the heterodimeric super-receptor function. Another unique characteristic of heterodimeric boosting in mGluR2/7 is that agonist binding in one subunit fully activates the receptor, in contrast to the partial activation seen with one-subunit agonism in mGluR2/2 homodimers[16]. We wondered if the triple mutant from mGluR4 to mGluR7 identity in the cysteine loop of mGluR4 would also confer this property onto the mGluR2/4 heterodimer. We tested this in two ways: with a group II agonist that would preferentially bind to the mGluR2 subunit of the heterodimer and with a group III agonist that would preferentially bind to mGluR4 subunit of the heterodimer. To selectively ligand mGluR2,

we used 1 $\mu$M LY379268, a concentration that is ~350-fold higher than the EC50 for mGluR2/2, and 10-fold lower than a concentration that has no effect on mGluR4/4[35]. To selectively ligand mGluR4, we used 10 $\mu$M LSP4-2022, a concentration that is ~100-fold higher than the EC50 for mGluR4/4, and 10-fold lower than a concentration that has no effect on mGluR2/2[36]. We found that the Group II-selective agonist LY379268 produced an ~50% occupancy of the low FRET activated conformation in wildtype mGluR2/4 (Fig. 7a, black) but full occupancy in mGluR2/4(G131T, G137T, G140E) (Fig. 7a, red). Similarly, the Group III-selective agonist LSP4-2022 produced ~50% occupancy of the low FRET activated conformation in wildtype mGluR2/4 (Fig. 7b, black) but full occupancy in mGluR2/4(G131T, G137T, G140E) (Fig. 7b, red). The ability of these residues to confer onto mGluR4 the mGluR7-heterodimeric boost in apparent affinity, efficacy and full activation by single subunit ligation of mGluR7 points to a key role for the cysteine loop in group II/III heterodimer cooperativity.

## Discussion

Two defining features of mGluRs are large N-terminal extracellular domains (ECDs) and obligate dimerization. Understanding of how the ECD and dimerization regulate activation, ligand affinity, efficacy, and the potency of individual ligating events is limited. We investigated the molecular mechanisms that regulate these properties by harnessing the pioneering development by Pin and colleagues of inter-subunit FRET between donor and acceptor fluorophore pairs attached to N-terminal SNAP-tagged and CLIP-tagged receptors that make it possible to monitor the activation rearrangements of the LBDs[24]. We did this in full-length receptors using smFRET in order to obtain absolute measures of FRET and, thus, fractional occupancy of distinct functional states of single receptor dimers. We observed a great diversity across mGluR family members in the apparent affinity for

glutamate and the maximal occupancy of the activated conformation. Group III mGluRs are limited in the stability of the active conformation and, within group III, poorer ability of glutamate to stabilize the activated conformation was associated with lower apparent affinity. One feature that distinguishes group III mGluRs from the others, and that gives them a unique pharmacological profile for orthosteric ligands, is a binding pocket that has three substitutions, compared to other mGluRs. Remarkably, these differences contribute little to glutamate apparent affinity and efficacy in stabilizing the activated conformation of the receptor.

To identify the molecular determinants that set affinity and efficacy, we made chimeras between high affinity and efficacy group II member mGluR2 and low affinity and efficacy group III member mGluR7, and between mGluR7 and another group III member, which has higher affinity and efficacy, mGluR4. We find that the LBD has a dominant influence, but that the CRD also contributes, with little influence of the TMD. In the LBD, there are contributions from both the upper and lower lobes, with the biggest contribution coming from the lower lobe's dimer interface. Interestingly, both this interface and the CRD only come into contact between subunits in the active state[28–30,37]. This suggests that limited stability of the active conformation in group III mGluRs may arise from steric clashes or weak interactions at these interfaces.

A striking feature of mGluR7 is that, as a homodimer, it requires a very high concentration of glutamate to begin to enter the activated LBD conformation and, even in saturating glutamate, never exceeds ~5% occupancy of that conformation[16]. In contrast, the heterodimer mGluR2/7 begins to enter the activated conformation at a lower glutamate concentration than the homodimer of its high affinity subunit, mGluR2/2[16]. Moreover, mGluR2/7 is fully activated by liganding of only one subunit (either mGluR2 or mGluR7), whereas the mGluR2/2 homodimer, like the group I mGluR5/5 homodimer, is only weakly activated by liganding of one subunit[16,38]. This indicates that heterodimeric interaction between mGluR2 and mGluR7 is favorable for stabilizing the active conformation. We find here that this super-receptor property of mGluR2/7 extends to another II/III heterodimer: mGluR2/4, but in a weaker form. We examine chimeras between mGluR4 and mGluR7 in effort to identify the determinant of the unique mGluR2/7 heteromeric boost. We find that the cysteine loop, which forms an inter-subunit disulfide bond between upper lobes of the LBDs, is a key determinant that confers strong cooperativity in the mGluR2/7 heterodimer. Substitution of three glycine residues in the mGluR4 cysteine loop with their mGluR7 counterparts is sufficient to transfer to mGluR4 the super-receptor properties of mGluR7. We propose that the glycine-rich nature of the mGluR4 cysteine loop renders it more flexible than the mGluR7 loop, reducing its ability to cooperatively interact with the loop on the partner subunit. Despite a surge in the determination of full-length mGluR homodimer and heterodimer structures[28–30,37,39], the high degree of similarity across these structures poses an obstacle to identifying molecular determinants of efficacy from structures alone[40,41]. Here, by employing chimeric swaps of the cysteine loop, which exhibits poor density in cryo-electron microscopy, we identify essential dimer interface interactions that would not have been identified via structural analysis alone. It is not known what interactions these three residues make, but one possible explanation is that the glycine-rich nature of the mGluR4 cysteine loop renders it more flexible than the mGluR7 loop, reducing its ability to cooperatively interact with the loop on the partner subunit.

In summary, cooperative interactions between subunits in the mGluR dimer tune activation via negative cooperativity in Group III homodimers and positive cooperativity in the mGluR2/7 heterodimer. Three molecular determinants contribute to this subunit interaction. The LBD lower lobe dimer interface and CRDs limit occupancy of the activated conformation in Group III members and the cysteine loop that forms a dimer bridge between LBD upper lobes is key to the

ability of mGluR7 to form a heterodimeric super-receptor with mGluR2.

Physiologically, the synaptic cleft experiences high but brief glutamate transients. Stabilization of the active state in heterodimers like mGluR2/7 may allow for more efficient entry into and residence in the active state during these transients. Consistent with this notion, the activation rates of the mGluR2/7 and mGluR3/7 heterodimers have been shown to be faster than those of the mGluR2/2 and mGluR3/3 homodimers[42]. Our observation that dimer interactions allosterically define the properties of agonist sensitivity and efficacy provides a paradigm for thinking about how mGluRs are tuned functionally. Incomplete occupancy of the active conformation due to unfavorable active state subunit interactions in the homodimers of the largest group of mGluRs provides headroom for positive modulation. Such modulation could arise from regulated expression, transport or co-assembly with heterodimeric partners, potentially by physiological ligands, or by interaction with other classes of proteins at specific cellular locations, such as synapses, as shown for ELFN1[23,43]. The dimer interfaces that we identify here hold promise as targets for allosteric drugs that alter the stability of distinct functional states. Such agents should be easier to make drugs that are more selective than those that target the conserved orthosteric site.

## Methods

### Cell culture and transfection

HEK293T cells were cultured in DMEM with 5% FBS on poly-L-lysine-coated glass coverslips. HEK293T cells were obtained from the UC Berkeley MCB tissue culture facility, authenticated by DDC Medical, and tested negative for mycoplasma contamination. Previously described HA–SNAP and Flag–CLIP-tagged rat mGluR cDNA were generously provided by J. P. Pin. DNA plasmids were transfected into cells using lipofectamine 2000 (Thermo Fisher). For FRET experiments, cells were transfected with SNAP and CLIP-tagged constructs at a ratio of 1:2 with 0.3 mg of SNAP–mGluR DNA per well.

### FRET dye labeling of SNAP- and CLIP- tagged mGluRs

Approximately 24–48 h after transfection, cells were labeled while attached to poly-L-lysine-coated coverslips. Culture media was removed and coverslips were washed and transferred to extracellular solution containing (in mM): 135 NaCl, 5.4 KCl, 2 CaCl$_2$, 1 MgCl$_2$, 10 HEPES, pH 7.4. Cells were labeled at 37 °C with one or two SNAP-reactive (benzylguanine, BG) dyes at 1.5 μM for 45 min, and then, if a CLIP-tagged mGluR was used, they were washed and labeled with a CLIP-reactive (benzylcytosine, BC) dye at 3 μM for 45 min. For most of the experiments DY-547 (NEB) was used as a donor and Alexa-647 (NEB) as an acceptor. Heterodimer experiments were labeled using LD-655 as the donor and LD-655 as the acceptor (Lumidyne). The fluorophores were diluted in extracellular solution and coverslips were washed in between labeling with donor and acceptor.

### smPull receptor isolation and surface display

To inhibit nonspecific protein adsorption, flow cells for single-molecule experiments were prepared as previously described[12] using mPEG (Laysan Bio) passivated glass coverslips (VWR) and doped with biotin PEG16. Before each experiment, coverslips were incubated with NeutrAvidin (Thermo), followed by 10 nM biotinylated secondary antibody (donkey anti-rabbit, Jackson ImmunoResearch). For receptor immunopurification, 10 nM anti-mGluR2 primary antibody (Abcam, ab150387) or 10 nM anti-mGluR7 antibody (Abcam, ab53705), or 15 nM anti-HA antibody (Abcam, ab26228) was incubated in the chamber (Fig. 1e). Between each conjugation step, the chambers were flushed to remove free reagents. The antibody dilutions and washes were done in T50 buffer (50 mM NaCl, 10 mM Tris, pH 7.5). For single-molecule experiments, fresh cells expressing tagged mGluR constructs were labeled, as described above, and recovered from coverslips by

incubating with $Ca^{2+}$ free PBS buffer for 5–10 min followed by gentle pipetting. Cells were then pelleted by spinning at 5000 g for 5 min and lysed in lysis buffer consisting of 150 mM NaCl, 1 mM EDTA, protease inhibitor cocktail (Thermo Scientific) and 1.2% IGEPAL (Sigma). Following 1 h incubation at 4 °C, lysate was centrifuged at 16,000 g for 20 min. and the supernatant was collected and kept on ice. To achieve sparse immobilization of labeled receptors on the surface, the cell lysate was diluted (ranging from 5× to 50× dilution depending on the expression and labeling efficiency) and applied to coverslips. After achieving optimum surface immobilization (~400 molecules in a 2000 $\mu m^2$ imaging area), unbound receptors were washed out of the flow chamber and the flow cells were then washed extensively (up to 50× the cell volume).

## smFRET measurements

Receptors were imaged for smFRET in imaging buffer consisting of (in mM) 3 Trolox, 120 KCl, 29 NaCl, 2 $CaCl_2$, 1 $MgCl_2$, 50 HEPES, 0.04% IGEPAL and an oxygen scavenging system (0.8% dextrose, 0.8 mg $ml^{-1}$ glucose oxidase, and 0.02 mg $ml^{-1}$ catalase), pH 7.4. Reagents were purchased from Sigma and were all UltraPure grade (purity >99.99%). All buffers were made in UltraPure distilled water (Invitrogen). For the experiments done in the absence of $Ca^{2+}$, 10 mM EGTA and 1 mM EDTA were added to the imaging buffer. Catalase was diluted in T50 buffer and passed through a spin column 3× (BioRad). Samples were imaged with a 1.65 na X60 objective (Olympus) on a total internal reflection fluorescence microscope with 100 ms time resolution unless stated otherwise. Lasers at 532 nm (Cobolt) and 632 nm (Melles Griot) were used for donor and acceptor excitation, respectively. FRET efficiency was calculated as $(I_A - 0.1 I_D)/(I_D + I_A)$, in which $I_D$ and $I_A$ are the donor and acceptor intensity, respectively, after back-ground subtraction. Imaging was with 100 ms acquisition time (10 Hz) with a Photometrics Prime 95B cMOS camera using Lumidyne LD555 as donor and Lumidyne LD655 as acceptor (Förster radius ~52 Å). Dyes were conjugated to benzyguanine and benzylecytosine to allow for labeling of SNAP and CLIP proteins, respectively.

## smFRET data analysis

Single-molecule intensity traces showing single-donor and single-acceptor photobleaching with a stable total intensity for longer than 5 s were collected (20–30% of total molecules per imaging area). Movies were analyzed using SPARTAN analysis software[44]. Individual traces were smoothed using a nonlinear filter[45] with the following filter parameters: window = 2, M = 2 and P = 15. Each experiment was performed ≥4 times to ensure reproducibility. smFRET histograms were compiled from ≥100 molecules per condition. (100 ms time resolution). Error bars in the histograms represent the standard error from ≥4 independent movies. To ensure that traces of different lengths contribute equally, histograms from individual traces were normalized to one before compiling. Active state occupancy was calculated as a percentage by dividing values at the active state FRET peak by the value of the same sample at the inactive state FRET peak.

## Reporting summary

Further information on research design is available in the Nature Portfolio Reporting Summary linked to this article.

## Data availability

The data that support this study are present within the figure and supporting information files. Source data are provided in this paper.

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

## Acknowledgements

We thank all members of the Isacoff Lab for their helpful comments. Support was provided by the National Institutes of Health (RO1NS119826 and 1RF1MH123246 to E.Y.I. and K99GM148823 to N.L.) and a UC Berkeley Miller Postdoctoral Fellowship (to N.L.). E.Y.I. is a Weill Neurohub Investigator.

## Author contributions

C.H. and E.Y.I conceived the project. C.H., N.L. and E.Y.I. wrote the manuscript. C.H. performed the experiments and analysis. N.L. performed the structural modeling. Z.F. helped with molecular biology.

## Competing interests

The authors declare no competing interest.
