## [Peer Review File · Nature Communications]

Homo- and hetero-dimeric subunit interactions set affinity and efficacy in metabotropic glutamate receptorsReviewers' Comments:

Reviewer #1:

Remarks to the Author:

The work by Habrian et al. is a single molecule FRET study of the steady state conformational changes in mGluR homo- (and to a lesser extent hetero-) dimers upon glutamate binding.

The authors aim to address the question of what exactly drives the higher fractional occupancy of the active state in mGluR2 (group II) as opposed to mGluR7 (group III). The authors work with chimaeras where the Ligand Binding Domain (LBD), Cystein rich domain (CRD) or Transmembrane domain (TMD) of such two receptors are pairwise exchanged, pointing to the LBD key role in modulating the higher population of the active conformational state upon glutamate binding,

Then the authors 'zoom' in to the specific region of the LBD that mediates the reduced population of the active conformation in mGluR3, highlighting the key role of the lower LBD lobe, which is further associated to its structural role as a dimeric interface. So, dimerization rather than residues differences at the protomer level, is the key modulator of the proportion of receptors that manage to get into an active conformation upon Glutamate binding. The role of this interface is then explored monitoring heterodimeric interactions (e.g. mGluR2/7), observing that they can boost the active conformation upon ligand binding, narrowing this effect down to the cystein loop.

The work provides a fascinating insight into the structural rearrangement of this family of receptors in vitro. The experiments are clean and provide a simple and elegant readout in terms of the relative population of the active/inactive states, associated respectively to low/high FRET conditions.

There are, however, a couple of major concerns that arise from reading the manuscript as it stands now:

1. The authors discuss from the beginning about how their data allow making statements about affinity and efficacy, separately. It is unclear how this is possible considering that their experiments provide a single readout: population of the active conformation vs inactive conformation. While the authors define 'efficacy' in relation to the number of molecules found in the active conformation (albeit, intuitively, one would expect efficacy to be associated to some form of downstream signalling readout), it is then unclear how affinity is measured, considering that ligands are not tagged.

A clarification of this aspect, or a removal of any comment about affinity, is in my opinion important in a revised version of the manuscript.

2. The role of heterodimers, explored in Figures 6 and 7, needs to be better explained in relation to the previous measurements. What is the rationale for switching to studying hetero-dimers in Figure 6? Wasn't the question at hand about why mGluR2 homodimers function better (i.e. more active conformation for equal Glutamate concentration) than mGluR7 homodimers?

3. Could the authors use transfected HEK cells to conduct actual affinity measurements with the chimaeras they use in Figure 2, as well as monitoring downstream response?

4. The manuscript needs to be thoroughly revised, as there are references to non-existing figures, such as Figure 1e.

Minor aspects include graphs axes not matching, or misaligned, making harder to compare the distribution of FRET efficiencies across different conditions (e.g. Figures 3, 5, and 6).

Reviewer #2:

Remarks to the Author:

The manuscript entitled 'homo-hetero-dimeric subunit interactions set affinity and efficacy in metabotropic glutamate receptors' by Habrian et al. is a work following the previous studies published in Nature (Vafabakhsh R et al., Nature 2015) and Nature Communications (Habrian C et al., 2019). Using the single-molecule FRET technique, the authors examined the conformational changes induced

by glutamate in group II/III homo and hetero dimeric metabotropic glutamate receptors. By creating mGluRs chimeras with swapping domains or mutations, the authors identified structural elements contributing to ligand efficacy, i.e., the lower lobe of the ligand binding domain (LBD) and cysteine-rich domain (CRD). Moreover, mutational studies revealed key residues essential for boosts in ligand affinity and efficacy in heterodimeric mGluRs. Overall, the present work uncovered important mechanisms underlying the affinity and efficacy of homo- and heterodimeric mGluRs. However, the authors only presented the histograms and state occupancies of their smFRET data collected from mGluRs. They did not provide analyses on the conformational dynamics of mGluRs (for example, dwell time changes, dynamic properties reflected by donor-acceptor cross-correlation, or transition density maps) and their dependences on ligands and swapping domains. Moreover, I felt the current studies could be further improved by utilizing the available cryo-EM structures of mGluRs to identify key residues mediating the subunit interactions using mutational analysis. In addition, there are many critical mistakes or caveats in the manuscript, including (1) Fig. 1e is missing, and legends of supplementary Fig 1 were misplaced; (2) The numbers of movies and traces to generate each histogram are not included; (3) Statistics to calculate occupancies of low and high FRET states were not provided; (4) Many figure legends do not contain sufficient information to interpret corresponding experimental results clearly.

Major comments:

1. The authors only briefly mentioned the trace number ≥ 100 from ≥ 4 movies. For all smFRET histogram data, the authors should include the number of movies and traces used to generate histograms and explain how s.e.m. was calculated, which are important to evaluate the quality of the smFRET data.
2. The authors did not describe how FRET state occupancies were calculated from each FRET histogram. I suggest the authors include a table of occupancies of different mGluRs under ligand conditions, so readers can easily follow or appreciate the changes in FRET state occupancies induced by glutamate and the effects of domain swap or mutations.
3. In the legends of Fig 1, the maximum activation was incorrectly defined. It is more likely the occupancies of the low FRET state rather than the ratio of low FRET peak/high FRET peak, as stated by the authors in the legends of Fig 1.
4. There are numerous mistakes in the supplementary Fig 1. Panel a's legends are missing; the legends of (a) should be for panel b, and (b) should be for panel c. The supplementary Fig 1d could be better illustrated with available Cryo-EM structures or, at the minimum, with cartoons by labeling LBD and CRDS domains clearly, so the readers can easily appreciate the idea that LBD rearrangements induce changes in CRDs and TMDs.
5. Line 173-176, the authors stated that "... but differ greatly in apparent affinity (by $\sim 10,000$ -fold) and efficacy (by ~ 12 -fold) (Fig. 1b, c, e)". However, it is unclear where these data came from, and Fig 1e is missing.
6. Determining the ligand efficacies of chimera mGlu2/7 is very interesting. I would recommend the authors perform more careful titration assays, especially on 2-2-7 and 2-7-7 chimeras and provide the half activation glutamate concentrations of different chimeras so the contribution of LBD, CRD and TMDs to the ligand efficacies of mGluRs can be quantitatively determined.
7. The results in Fig. 4 identified interface residues in the LBD lower lobe of the mGluR4. Based on the available mGluR4 cryo-EM structures, it would be more interesting if the authors could further nail down the interacting residue pairs.
8. In Fig. 5b, the authors claimed that mGluR4 with mGluR2 CRD has higher ligand efficacy, but it is hard to tell from the figures. I would like to suggest that authors include a table of FRET occupancy data from different mGluR constructs under different ligand conditions, ideally with error estimations.
9. The studies in Fig 6 and 7 are very interesting. However, it seems that 3 mutations shifted the center of the high FRET peak significantly. Do the authors have any explanations for the shift? As reported in the manuscript, with 3 mGluR4 to mGluR7 mutations, ligand activation was promoted significantly in mGluR2/4 homodimers, including the ability of agonist activation of the entire heterodimer at only one subunit. But, if the authors could show that mGluR7 to mGluR4 mutations (i.e., T131G, T137G, E140G) can attenuate ligand activation in mGluR2/7 heterodimers, which would

be more compelling evidence to support their proposed mechanisms (i.e., interfacing controlling of ligand efficacy in mGluRs heterodimers).

We thank the reviewers for their helpful comments. We have addressed each of the points and added new experiments (Figure 6d) to address one of the major points and a new version of an explanatory cartoon (Suppl. Fig. 1d) that illustrates the way that smFRET reports on the conformational changes of activation, as summarized below.

Reviewer #1 (Remarks to the Author):

The work by Habrian et al. is a single molecule FRET study of the steady state conformational changes in mGluR homo- (and to a lesser extent hetero-) dimers upon glutamate binding.

The authors aim to address the question of what exactly drives the higher fractional occupancy of the active state in mGluR2 (group II) as opposed to mGluR7 (group III). The authors work with chimaeras where the Ligand Binding Domain (LBD), Cystein rich domain (CRD) or Transmembrane domain (TMD) of such two receptors are pairwise exchanged, pointing to the LBD key role in modulating the higher population of the active conformational state upon glutamate binding,

Then the authors 'zoom'in to the specific region of the LBD that mediates the reduced population of the active conformation in mGluR3, highlighting the key role of the lower LBD lobe, which is further associated to its structural role as a dimeric interface. So, dimerization rather than residues differences at the protomer level, is the key modulator of the proportion of receptors that manage to get into an active conformation upon Glutamate binding. The role of this interface is then explored monitoring heterodimeric interactions (e.g. mGluR2/7), observing that they can boost the active conformation upon ligand binding, narrowing this effect down to the cystein loop.

The work provides a fascinating insight into the structural rearrangement of this family of receptors in vitro. The experiments are clean and provide a simple and elegant readout in terms of the relative population of the active/inactive states, associated respectively to low/high FRET conditions.

There are, however, a couple of major concerns that arise from reading the manuscript as it stands now:
1. The authors discuss from the beginning about how their data allow making statements about affinity and efficacy, separately. It is unclear how this is possible considering that their experiments provide a single readout: population of the active conformation vs inactive conformation.

While the authors define 'efficacy' in relation to the number of molecules found in the active conformation (albeit, intuitively, one would expect efficacy to be associated to some form of downstream signalling readout), it is then unclear how affinity is measured, considering that ligands are not tagged. A clarification of this aspect, or a removal of any comment about affinity, is in my opinion important in a revised version of the manuscript.

Indeed, we do not measure ligand binding. We therefore refer to "apparent affinity" based on the EC50 for transition from the high FRET resting state to the low FRET activated state. In some places we had left out the "apparently" for readability. "Apparently" has now been added in throughout to emphasize that we are measuring the effect of ligand on receptor conformation and not the binding of ligand.

2. The role of heterodimers, explored in Figures 6 and 7, needs to be better explained in relation to the previous measurements. What is the rationale for switching to studying hetero-dimers in Figure 6? Wasn't the question at hand about why mGluR2 homodimers function better (i.e. more active conformation for equal Glutamate concentration) than mGluR7 homodimers?

We address two related questions about cooperative interactions between subunits in the mGluR dimer that tune activation: negative cooperativity in Group III homodimers and positive cooperativity in the mGluR2/7 heterodimer. We ask what limits the ability of group III homodimers to occupy the activated state and what makes the mGluR2/7 (but not mGluR2/4) into a super-receptor. We discover three contributing molecular determinants of subunit interaction:

- a) **LBD lower lobe dimer interface.** This interface limits occupancy of the activated conformation in Group III members (**Figs. 1-4**).
- b) **CRDs.** The CRDs Group III member mGluR7 (**Fig. 2**) and of mGluR4 (**Fig. 5**) limit occupancy of the activated conformation in comparison to Group II member mGluR2.
- c) **Cysteine loop that forms a dimer bridge between LBD upper lobes.** Three cysteine loop residues are key to the ability of Group III member mGluR7 to form a hetero-dimeric super-receptor with mGluR2 (**Figs. 6-7**).

For clarification, we have added this to the penultimate paragraph of the Discussion.

3. Could the authors use transfected HEK cells to conduct actual affinity measurements with the chimaeras they use in Figure 2, as well as monitoring downstream response?

Unfortunately, we are not equipped to do ligand binding studies. And we have agreed with Reviewer-1 Comment-1 and reduced emphasis on affinity to focus on efficacy.

We should add that activation of the downstream response (say using the GIRK channel as an effector) would be even further downstream of ligand binding than the conformational change in the receptor. And while it would reflect affinity in an indirect way it could not show efficacy, which demands the reference of a full agonist (which we do not have for the Group III members) in any ensemble (macroscopic) recording. In contrast, our single molecule analysis has the great advantage of providing an absolute measure of occupancy of the activated state under saturating ligand, thus directly showing efficacy. We therefore rely on the conformational change in the LBD as the most proximal indicator of binding, which triggers a rearrangement from a low affinity open LBD resting state to a high affinity closed LBD activated state.

4. The manuscript needs to be thoroughly revised, as there are references to non-existing figures, such as Figure 1e.

We thank the reviewer for catching this and have done a thorough job of editing.

Minor aspects include graphs axes not matching, or misaligned, making harder to compare the distribution of FRET efficiencies across different conditions (e.g. Figures 3, 5, and 6).

We have adjusted the size of the FRET distribution plots to ensure that they are of identical size to ensure that they can be compared. To maximize use of the figure space and ease comparison between conditions, some histograms differ in y-axis based on the maximal height of a given sample's distribution.

Reviewer #2 (Remarks to the Author):

The manuscript entitled 'homo-hetero-dimeric subunit interactions set affinity and efficacy in metabotropic glutamate receptors' by Habrian et al. is a work following the previous studies published in Nature (Vafabakhsh R et al., Nature 2015) and Nature Communications (Harbrian C et al., 2019). Using the single-molecule FRET technique, the authors examined the conformational changes induced by glutamate in group II/III homo and hetero dimeric metabotropic glutamate receptors. By creating mGluRs chimeras with swapping domains or mutations, the authors identified structural elements contributing to ligand efficacy, i.e., the lower lobe of the ligand binding domain (LBD) and cysteine-rich domain (CRD). Moreover, mutational studies revealed key residues essential for boosts in ligand affinity and efficacy in heterodimeric mGluRs. Overall, the present work uncovered important mechanisms underlying the affinity and efficacy of homo- and heterodimeric mGluRs.

However, the authors only presented the histograms and state occupancies of their smFRET data collected from mGluRs. They did not provide analyses on the conformational dynamics of mGluRs (for example, dwell time changes, dynamic properties reflected by donor-acceptor cross-correlation, or transition density maps) and their dependences on ligands and swapping domains. Moreover, I felt the current studies could be further improved by utilizing the available cryo-EM structures of mGluRs to identify key residues mediating the subunit interactions using mutational analysis. In addition, there are many critical mistakes or caveats in the manuscript, including (1) Fig. 1e is missing, and legends of supplementary Fig 1 were misplaced; (2) The numbers of movies and traces to generate each histogram are not included; (3) Statistics to calculate occupancies of low and high FRET states were not provided; (4) Many figure legends do not contain sufficient information to interpret corresponding experimental results clearly.

Major comments:

1. The authors only briefly mentioned the trace number ≥ 100 from ≥ 4 movies. For all smFRET histogram data, the authors should include the number of movies and traces used to generate histograms and explain how s.e.m. was calculated, which are important to evaluate the quality of the smFRET data.

We have now added to the legends the number of traces and movies in each figure panel. An explanation of how s.e.m. is calculated has been added to Methods.

2. The authors did not describe how FRET state occupancies were calculated from each FRET histogram. I suggest the authors include a table of occupancies of different mGluRs under ligand conditions, so readers can easily follow or appreciate the changes in FRET state occupancies induced by glutamate and the effects of domain swap or mutations.

We have added to Methods a description of how FRET state occupancies were calculated for each FRET histogram. We have also included measures of active state occupancy in the legends alongside the number of traces as a percentage. The description of how these percentages were calculated is included in Methods.

3. In the legends of Fig 1, the maximum activation was incorrectly defined. It is more likely the occupancies of the low FRET state rather than the ratio of low FRET peak/high FRET peak, as stated by the authors in the legends of Fig 1.

We thank the reviewer for catching this error. We have now corrected the definition of occupancy of the low FRET state as maximal activation (low FRET peak / low FRET peak + high FRET peak).

4. There are numerous mistakes in the supplementary Fig 1. Panel a's legends are missing; the legends of (a) should be for panel b, and (b) should be for panel c. The supplementary Fig 1d could be better illustrated with available Cryo-EM structures or, at the minimum, with cartoons by labeling LBD and CRDS domains clearly, so the readers can easily appreciate the idea that LBD rearrangements induce changes in CRDs and TMDs.

We thank the reviewer for catching this error. We have added the correct panel Fig. S1a legend and relabeled the other panels. No one view of the active and inactive mGluR2 LBD structures conveys the movements due to the rotation of the LBDs relative to one another and as a result depicting this change through structures requires multiple views and becomes confusing. We therefore revised the cartoon version of Fig. S1d in which we now show the whole subunit in the resting and activated conformations and indicate how the N-terminal SNAP domain dyes change distance (and therefore FRET) in the activation rearrangement. The domains are illustrated as in Suppl. Fig. 1a, where domain labels are provided.

5. Line 173-176, the authors stated that ‘..., but differ greatly in apparent affinity (by ~10,000-fold) and efficacy (by ~12-fold) (Fig. 1b, c, e)’. However, it is unclear where these data came from, and Fig 1e is missing.

We have changes this to: “but differ greatly in efficacy (by ~12-fold) (**Fig. 1b, c**) and apparent affinity (by ~10,000-fold) (Habrian et al., 2019)” to indicate that our data here demonstrate the difference in efficacy and that the difference in affinity depends on a comparison to our earlier work. We have removed the callout to the missing Fig.1e, whose data is presented, instead, at the end of the legend to Fig. 1.

6. Determining the ligand efficacies of chimera mGlu2/7 is very interesting. I would recommend the authors perform more careful titration assays, especially on 2-2-7 and 2-7-7 chimeras and provide the half activation glutamate concentrations of different chimeras so the contribution of LBD, CRD and TMDs to the ligand efficacies of mGluRs can be quantitatively determined.

We agree that precise quantification of apparent affinity (EC50) from glutamate concentration – FRET relations would be interesting for the domain swap chimeras in Figure 2, as well as for the smaller chimeras presented in the figures that follow. However, since we agreed with the suggestion to remove emphasis on affinity and focus on efficacy we feel that the added delay to perform this large series of experiments is not warranted. The three ligand conditions in the figure are sufficient to justify the core interpretation that, at 10 μ M glu, 2-2-7 is more than half activated whereas 2-7-7 is less than half activated, i.e. a higher apparent affinity for 2-2-7. As regards efficacy, our prior work (Habrian et al., 2019) showed that, in mGluR7/7, 100 mM glu is saturating (i.e. no further increase at 250 mM), reaching 5-10% maximal activation. Compared to this, Figure 2 shows that 2-7-7 has much higher efficacy and 2-2-7 higher still.

7. The results in Fig. 4 identified interface residues in the LBD lower lobe of the mGluR4. Based on the available mGluR4 cryo-EM structures, it would be more interesting if the authors could further nail down the interacting residue pairs.

The mGluR4 structures show that the dimer interface involves a surface and not one residue pair or two. This is in keeping with our finding that we could not transfer properties in smaller chimeras that sub-divided the ~70 amino acid region of what we described as the minimal interface (**Supp. Fig. 3b, c and d**). We describe this on the top of page 8 at the end of the Results section entitled “Lower LBD interface a key determinant of group III mGluR low efficacy.”

8. In Fig. 5b, the authors claimed that mGluR4 with mGluR2 CRD has higher ligand efficacy, but it is hard to tell from the figures. I would like to suggest that authors include a table of FRET occupancy data from different mGluR constructs under different ligand conditions, ideally with error estimations.

We have added a table of FRET occupancy data to the figure legends throughout the paper.

9. The studies in Fig 6 and 7 are very interesting. However, it seems that 3 mutations shifted the center of the high FRET peak significantly. Do the authors have any explanations for the shift? As reported in the manuscript, with 3 mGluR4 to mGluR7 mutations, ligand activation was promoted significantly in mGluR2/4 homodimers, including the ability of agonist activation of the entire heterodimer at only one subunit. But, if the authors could show that mGluR7 to mGluR4 mutations (i.e., T131G, T137G, E140G) can attenuate ligand activation in mGluR2/7 heterodimers, which would be more compelling evidence to support their proposed mechanisms (i.e., interfacing controlling of ligand efficacy in mGluRs heterodimers).

We originally attempted to identify a motif whose transplantation would transfer heterodimeric super-receptor properties from mGluR7 to mGluR4. Our results showed that switching 3 cysteine loop residues of mGluR4 to the identity of those in mGluR7 endows mGluR4 with super-receptor behavior in the mGluR2/4 heterodimer. This demonstrates that these 3 cysteine loop residues are “sufficient” to mediate the strong positive cooperativity between mGluR2 and its Group III heterodimeric partner. We now followed the reviewer’s suggestion and added a new complementary experiment in which we made the reverse swap. We find that the transfer to mGluR7 of the GGG from mGluR4 reduces occupancy of the active low FRET conformation at both intermediate (10 μ M) and high (100 mM) glutamate (**new Fig. 6d**), indicating that these residues are “necessary” for super-receptor function.

As pointed out by the reviewer, compared to mGluR2/4 (**Fig. 6b**), the heterodimer of mGluR2 with an mGluR4 whose cysteine loop contains the 3-residue mutation to mGluR7 identities [mGluR2/4(G131T, G137T, G140E)] has a left-shifted high FRET distribution in zero glutamate (**Fig. 6c**). The left shift moves the zero glutamate FRET distribution well to the left of that seen in the mGluR2/2 and mGluR7/7 homodimers (**Fig. 1a,b**). As we showed earlier, the left shift of the mGluR2/7 zero glutamate FRET distribution reflects partial activation of mGluR2/7 in the Apo (glutamate-free) state, which can be reversed (shifted back to the right) by negative allosteric modulators but not by an orthosteric antagonist, indicating partial rotation without clamshell closure. Strikingly, compared to mGluR2/7(**Fig. 6a**), the reverse swap in mGluR7 [mGluR2/7(T131G,

T137G, E140G]] has the opposite effect: it right-shifts the high FRET distribution in zero glutamate (**new Fig. 6d**), consistent with the elimination of entry part-way into the activation pathway in zero glutamate. In other words, the identity of the 3 cysteine loop residues determines not only total efficacy when both subunits are liganded by glutamate (**Fig. 6, red symbols**) and efficacy due to agonist binding in only one subunit (**Fig. 7**), it also determines whether or not the receptor is primed for activation by spontaneous entry part-way into the activation pathway in zero glutamate (**Fig. 6, black symbols**). We have incorporated these points into the Discussion.

Reviewers' Comments:

Reviewer #2:

Remarks to the Author:

In the revised manuscript, the authors have sufficiently addressed the issues raised in the first review. I have no further comments on this work.